# Pushing the Limits of Narrow Precision Inferencing at Cloud Scale with Microsoft Floating Point

**Bita Rouhani** [*]
Microsoft

**Daniel Lo** [*]
Microsoft

**Ritchie Zhao**
Microsoft

**Ming Liu**
Microsoft

**Jeremy Fowers**
Microsoft

**Kalin Ovtcharov**
Microsoft

**Anna Vinogradsky** [†]
Caltech

**Sarah Massengill**
Microsoft

**Lita Yang**
Microsoft

**Ray Bittner**
Microsoft

**Alessandro Forin**
Microsoft

**Haishan Zhu**
Microsoft

**Taesik Na**
Microsoft

**Prerak Patel**
Microsoft

**Shuai Che**
Microsoft

**Lok Chand Koppaka**
Microsoft

**Xia Song**
Microsoft

**Subhojit Som**
Microsoft

**Kaustav Das**
Microsoft

**Saurabh Tiwary**
Microsoft

**Steve Reinhardt**
Microsoft

**Sitaram Lanka**
Microsoft

**Eric Chung**
Microsoft

**Doug Burger**
Microsoft

## Abstract

In this paper, we explore the limits of Microsoft Floating Point (MSFP), a new class of datatypes developed for production cloud-scale inferencing on custom hardware. Through the co-evolution of hardware design and algorithms, MSFP16 incurs $3\times$ lower cost compared to Bfloat16 and MSFP12 has $4\times$ lower cost compared to INT8 while delivering a comparable or better accuracy. MSFP incurs negligible impact to accuracy (<1%), requires no changes to the model topology, and is integrated with a mature cloud production pipeline. MSFP supports various classes of deep learning models including CNNs, RNNs, and Transformers without modification. Finally, we characterize the accuracy and implementation of MSFP and demonstrate its efficacy on a number of production scenarios, including models that power major online scenarios such as web search, question-answering, and image classification.

## 1 Introduction

Over the past few years, there has been an exponential growth in the size of deep neural networks (DNNs) to further push achievable accuracy [1, 2, 3]. With the diminishing of Moore's law, the arithmetic density that can fit on computing hardware plays an important role for large-scale inferencing. One key to increasing arithmetic density is the use of narrow bit-width datatypes.

Inferencing DNNs with narrow bit-width at a required service level agreement (i.e., accuracy and latency) requires a careful balance of dynamic range and hardware complexity. For instance, although narrow fixed-point datatypes incur a low hardware overhead, they lack a wide enough dynamic range. As such, the use of fixed-point arithmetic at large-scale is typically limited due to noticeable accuracy

---

[*]Equal contribution. Email correspondence to birouhan@microsoft.com
[†]work done while Anna Vinogradsky was an intern at Microsoft.

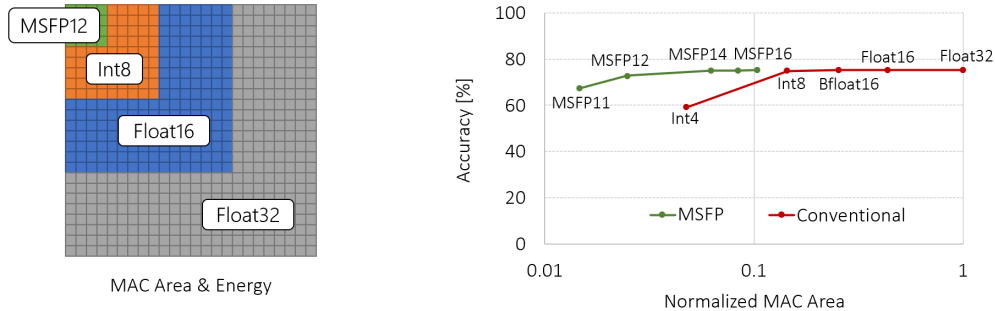

Figure 1: MSFP significantly improves upon previous datatypes in computational efficiency at each fixed level of accuracy. Left: relative area and energy cost of multiply-accumulate (MAC) using different datatypes on the same silicon. Right: ImageNet accuracy for ResNet-50 plotted versus normalized area cost. The area costs are normalized to Float32.

drops (>1%). Fixed-point datatypes also require a manual calibration process for each new benchmark. To address these inefficiencies, there is a rising interest in custom datatypes specifically designed for DNN workloads. Google deployed a custom datatype called Bfloat16 on its TPUs [4, 5] and NVIDIA recently announced a new datatype called TF32 available on its latest generation A100 GPUs [6]. Both Bfloat16 and TF32 represent a wide dynamic range which leads to close to zero accuracy drop when the DNN model is executed with these datatypes. However, while these datatypes have a lower hardware footprint compared to IEEE-compliant Float32, their overhead is still considered to be high for low-cost inference at scale. As a result, the industry seems to be converging to INT8 for inferencing which requires a careful model re-calibration to preserve accuracy. With the current setup, going below 8-bits almost always results in an accuracy drop.

This paper introduces Microsoft Floating Point (MSFP), a class of new datatypes for robust and low-cost DNN inference at scale. MSFP is a hardware/algorithm co-designed numerical format that enables an efficient realization of dot products (which are the building blocks of DNN workloads) on custom hardware while maintaining a high dynamic range ($[2^{-126}, 2^{127}]$). Figure 1 corroborates the accuracy-area trade-off of using different datatypes for serving ResNet50-ImageNet. MSFP outperforms existing datatypes in terms of area and energy cost when the model is held to a fixed accuracy. Variants of MSFP together form a new Pareto frontier for computational performance/$mm^2$ compared to a collection of competitive datatypes. In this paper, we quantify the Pareto frontier in terms of arithmetic density, the measure of how many dot products can be fit into $1\ mm^2$ of silicon on $16nm$ process. MSFP16 can be used as a drop-in replacement for Bfloat16 without any accuracy drop or requiring any re-calibration or hyper-parameter tuning. MSFP16 provides $2\times$ memory saving and $2.8\times$ higher arithmetic density compared to Bfloat16. We further built a fully automated fine-tuning pipeline to enable serving DNNs at even a lower cost while preserving the accuracy. With moderate model fine-tuning, MSFP can provide $4\times$ higher arithmetic density compared to INT8 industry-standard datatype for inference while delivering comparable accuracy.

MSFP is deployed at large-scale production for industry web services and has been successfully validated on over a dozen proprietary and open-sourced benchmarks. Extensive evaluations of a variety of computer vision and natural language processing models demonstrate the robustness and generality of the MSFP format. In summary, we make the following contributions:

- We propose Microsoft floating point, a hardware/algorithm co-designed numerical datatype for DNN workloads that can achieve the same accuracy level of existing datatypes at a fraction of the area and power cost on custom silicon.

- We build a low-friction production pipeline for serving pre-trained DNN models. MSFP preserves model accuracy at ultra-narrow bit-width with as few as three mantissa bits (i.e., MSFP12) with minimal fine-tuning. All conversions of weights and activations to MSFP format are handled in-situ on custom hardware.

- We perform extensive evaluations of MSFP on various CNN, RNN, and Transformer models. Deploying DNN models using the MSFP datatype leads to a new state-of-the-art Pareto frontier between accuracy and computational cost.

## 2 Microsoft Floating Point

MSFP is a hardware/algorithm co-designed numerical format for DNN workloads. Here we build on IEEE-compliant formats to introduce the structure of MSFP and elaborate on its functionality.

IEEE floating point formats include one sign bit $s$, a number of exponent bits $e$, and a number of significand or mantissa bits $m$. Float16, for instance, consists of $1$ sign bit, $5$ exponent bits, and $11$ mantissa bits (10 of which are explicitly stored). The resulting value can be decoded as $x = (-1)^s * 2^{e'} * m$ where $e'$ is set to $e - 15$ to adjust for the encoded bias. MSFP has a similar structure with one main difference. Instead of assigning a private exponent to each element of a tensor, MSFP relies on using a shared exponent among some number of values. For instance, for a vector of elements, the floating point representation is:

$$[(-1)^{s_0}\ 2^{e_0}\ m_0\ ,\ (-1)^{s_1}\ 2^{e_1}\ m_1\ ,\ ...\ ,(-1)^{s_{n-1}}\ 2^{e_{n-1}}\ m_{n-1}\ ],$$

the MSFP representation is:

$$2^{e_{shared}}\ \left[(-1)^{s_0}\ m'_0\ ,\ (-1)^{s_1}\ m'_1,\ ...\ ,(-1)^{s_{n-1}}\ m'_{n-1}\ \right].$$

The number of elements sharing one exponent is referred to as the bounding-box size. The shared exponent can be any value that is representative of the range of elements in each bounding-box. We use the $maximum$ exponent in our setting to best represent the outliers in each bounding-box. However, other approaches could be used such as taking an average or percentile value.

By associating each element in the bounding-box with the max exponent in the box, each mantissa term $m_i$ must be adjusted by shifting it to the right by the difference between $e_{max}$ and $e_i$. The term $m'_i$ is defined as $m_i \gg (e_{shared} - e_i)$, where $\gg$ is the right-shift operator. As $e_{max} - e_i$ increases, the right shift will truncate away more of the least-significant bits in the mantissa. Thus, similar to fixed-point, MSFP is affected by extreme outlier values. However, because each bounding box defines its own dynamic range, an outlier's effect would be limited to the bounding-box in which it occurs. In MSFP, zero is represented by having all mantissa bits being $0$ for a given value (shared exponent can be any value). MSFP mantissas do not have an implicit leading bit and all mantissa bits are explicitly represented.

The key insight behind MSFP is to strike a compromise between the dynamic range of floating point and the hardware efficiency of fixed-point. Floating point uses an exponent for every element, while fixed-point (with scaling) uses one exponent for all elements. In contrast, MSFP uses one exponent for each $n$ elements, and is able to approach the benefits of both formats.

**Computing with the MSFP format.** MSFP is selectively applied to performance-critical components of a model that exhaust computation resources and memory bandwidth. Dot product is one of the core operations involved in DNN inference, being the basic operation underlying both convolutional and fully-connected layers. Suppose we have two $n$-dimensional row-vectors $\overrightarrow{x_0}$ and $\overrightarrow{x_1}$ in MSFP format with shared exponents $2^{e_0}$ and $2^{e_1}$, respectively. The dot product of these vectors takes the form:

$$
\begin{aligned}
\overrightarrow{x_0}.\overrightarrow{x_1}^T =\ & 2^{e_0} \left[(-1)^{s_{0,0}}\ m'_{0,0}\ ,\ (-1)^{s_{0,1}}\ m'_{0,1}\ ,\ ...\ ,(-1)^{s_{0,n-1}}\ m'_{0,n-1}\ \right]. \\
& 2^{e_1} \left[(-1)^{s_{1,0}}\ m'_{1,0}\ ,\ (-1)^{s_{1,1}}\ m'_{1,1}\ ,\ ...\ ,(-1)^{s_{1,n-1}}\ m'_{1,n-1}\ \right]^T \\
=\ & 2^{e_0+e_1} \sum_{i=0}^{n-1} \left( (-1)^{s_{0,i} \oplus s_{1,i}}\ m'_{0,i} * m'_{1,i} \right),
\end{aligned}
$$

where $\oplus$ is XOR operation and $T$ stands for transposition. As shown, the dot product in MSFP format consists of a single fixed-point addition of exponents, $n$ fixed-point multiplications of mantissas, and $n - 1$ fixed-point additions of mantissa products. Here, $n$, the length of the dot product, coincides with the bounding-box size. However, this does not always need to hold. A large dot product can be built by summing the results of smaller bounding-box sized dot products. The overhead of MSFP compared to pure fixed-point is precisely the hardware needed to handle the shared exponent, and this overhead is amortized over the bounding-box size.

Figure 2 shows the high level overview of a systolic tensor core architecture (left) which computes long dot products using multiple bounding-box length MSFP dot product units (right). Within MSFP dot product, the multipliers and adders are all fixed-point. Thus for the same number of mantissa bits, MSFP has a significantly lower circuit footprint compared to IEEE floating point. By truncating the number of bits assigned to each mantissa, the circuit area can be reduced even further.

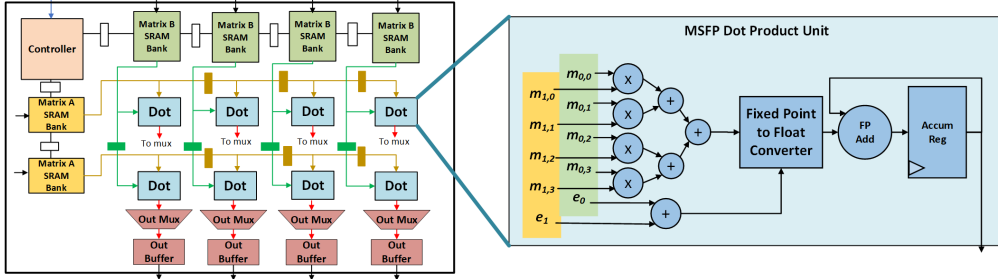

Figure 2: Systolic tensor core architecture containing multiple MSFP dot product units. Because each bounding-box has a single shared exponent, the math per-element can be done in fixed-point and the cost of exponent handling is amortized over $n$ (bounding box size). MSFP provides $3\times$ and $4\times$ higher MAC density compared to industry standard Bfloat16 and Int8, respectively.

Table 1 summarizes the dot product density and memory savings with MSFP format for different numbers of mantissa bits. The bounding-box size here is 16. Throughout the paper, we refer to different MSFP configurations as MSFP$N$ (e.g., MSFP12). The number listed is the sum of the bit-width assigned to sign, mantissa, and shared exponent. For the rest of the paper, MSFP will follow a sign-magnitude format and has an 8-bit shared exponent. The default bounding-box size is 16 unless explicitly mentioned otherwise. We will show in Section 4 that a bounding-box size of 16 provides a reasonable balance between inference cost and accuracy across various DNN benchmarks. MSFP12's MAC circuit size is smaller even than Int4. MSFP uses a sign-magnitude mantissa format, which costs less area and energy compared to conventional two's complement integer format (assuming equal mantissa bit-width).

Table 1: MSFP versus other commonly used numerical formats for DNN inference. Memory and MAC density of various formats are normalized to Float32. In this table, MSFP is assumed to have 8-bit exponent and a bounding-box size of 16. The results listed here are based on topographical synthesis results using TSMC $16nm$ FF+.

| | Float32 | Float16 | Bfloat16 | MSFP16 | MSFP15 | MSFP14 | MSFP13 | MSFP12 | MSFP11 | Int8 | Int4 |
|---|---|---|---|---|---|---|---|---|---|---|---|
| **Memory Density** | $1.0\times$ | $2.0\times$ | $2.0\times$ | $3.8\times$ | $4.3\times$ | $4.9\times$ | $5.8\times$ | $7.1\times$ | $9.1\times$ | $4.0\times$ | $8.0\times$ |
| **MAC Density** | $1.0\times$ | $1.8\times$ | $3.1\times$ | $8.8\times$ | $10.8\times$ | $13.9\times$ | $18.3\times$ | $31.9\times$ | $50.9\times$ | $7.7\times$ | $20.9\times$ |

## 3   MSFP configuration

In this section, we first discuss the effects of different configuration settings for the MSFP datatype. We conclude the section with the MSFP quantization pipeline for DNN workloads.

**Bounding-box size.**    The granularity of values which share an exponent (bounding-box size) is an important factor for both model accuracy and hardware cost. Sharing an exponent among fewer values improves the encoding efficiency and sharing exponents amongst a larger number of values helps keep hardware costs down. Figure 3 illustrates the Kullback-Leibler (KL) divergence between MSFP encoding and Float32 encoding for various bounding-box sizes and mantissa bit-widths in a layer sampled from ResNet-50. KL divergence between two encoding format $P$ and $Q$ is defined as $KL(P \parallel Q) = \sum_{x \in \mathcal{X}} P(x) \log\left(\frac{P(x)}{Q(x)}\right)$. Intuitively, a lower KL divergence translates to a lower discrepancy between data distributions before and after quantization; thus resulting in better accuracy. As shown in Figure 3, with fewer mantissa bits, smaller bounding-box sizes are required to keep the quantization error under control. In practice, we found a bounding-box size of 16-128 to be effective in preserving the accuracy while incurring a moderate hardware cost.

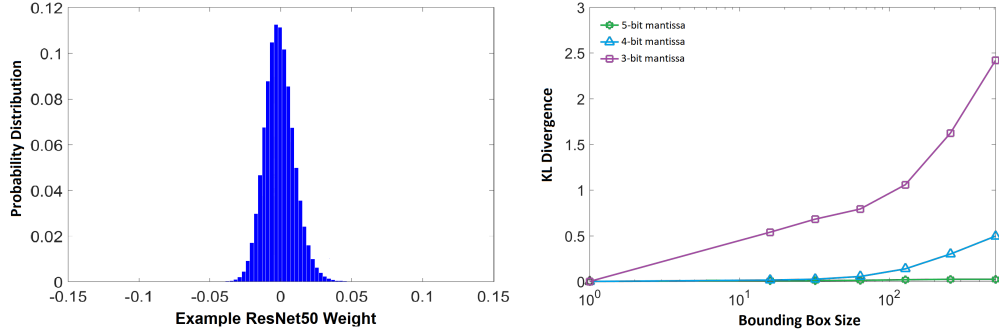

Figure 3: Effect of bounding-box size on the quantization error. (left) Histogram of pre-trained Float32 values in a sample layer of ResNet-50 model. (right) KL divergence of MSFP configuration and Float32 counterpart as a function of bounding-box size for different mantissa bit-width. Increasing the bounding-box size can reduce the hardware complexity at a cost a lower encoding efficiency.

**Bounding-box shape.** The shape of a bounding-box impacts the computing cost and ultimate model accuracy. While clustering similar magnitude values to create pertinent bounding-boxes will reduce quantization error, tracking exponents for arbitrary bounding-box shapes is expensive.

Figure 4 presents several hardware-friendly options to partition matrices into bounding-boxes. Consider a right-hand matrix multiply $y = xW$, where $x$ is a row-vector and $W$ is a matrix. The simplest approach is to treat the entire matrix $W$ as a single bounding-box. This is a common approach used in prior works [7]. Even though such a coarse-grained approach incurs a low hardware overhead, this can lead to severe accuracy loss due to outliers and needs careful re-calibration per benchmark. In a right-hand matrix multiply, dot products are performed between the row vector $x$ and columns of the matrix $W$. Another natural boundary is to treat each column of the matrix $W$ as a separate bounding-box. By aligning bounding-boxes to the columns of the matrix, all dot products are still between a pair of bounding-boxes which can be calculated using fixed-point arithmetic. Large matrices typically are broken down into small tiles that fit into limited hardware resources of a kernel. Thus, one may more effectively split the computation into finer-grained regions that align with hardware tiles (see the right-most images in Figure 4). We chose to work with tile-based partitioning to obtain a balance between accuracy and hardware cost. A similar strategy is applied to convolution layers by sharing the exponent along the channel depth.

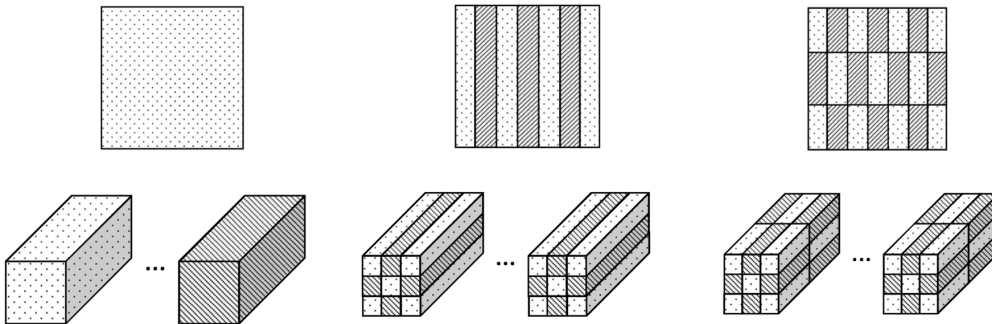

Figure 4: Hardware-friendly bounding-box shapes for (top row) fully-connected layers and (bottom row) convolution filters. We use tile-based partitioning (right most configuration) in our experiments. All conversions happens in-situ on hardware without requiring the user to perform any pre-processing.

**Encoding efficiency.** The encoding efficiency of MSFP depends on two main factors: bit-width and bounding-box size. Figure 5 demonstrates the expected value of Quantized Noise to Signal Ratio (QNSR) as a function of bit-width for MSFP format. In particular, the expected value of QNSR is defined as $E(\frac{Q(x)-x}{x})$, where $x$ is a random entry of random tensor $X$ and $Q(x)$ is the corresponding quantized value. To measure MSFP encoding efficiency, we considered thousands of random tensors

sampled from parameters of different neural networks and report the average acquired QNSRs (for output of a dot-product) in Figure 5. The overall variation across different tensors are shown using the error bars. As demonstrated, adding 1 bit to the mantissa decreases QNSR by almost $3.2dB$. Figure 5 further illustrates MSFP encoding efficiency as a function of bounding-box size. In this experiment, we used 6 bits of mantissa, 1 sign bit, and 8 bits of the shared exponent for conversion to MSFP format (i.e., MSFP15). As shown, doubling the block-size increases the QNSR by approximately $0.52dB$. The same trend is observed with other mantissa bit-widths.

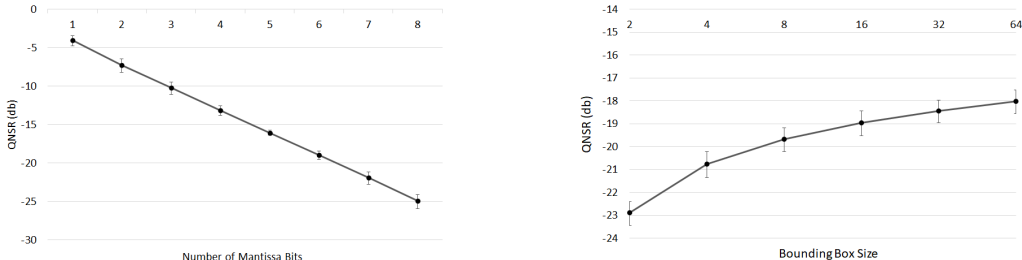

Figure 5: Encoding efficiency of MSFP format as a function of (left) number of mantissa bits and (right) bounding-box size. The reported QNSRs are averaged over thousands of random tensors.

**Quantization pipeline and deployment** In this paper, we focus on efficient inference of pre-trained deep neural networks. Although quantization-aware training of DNNs (e.g., with injected quantization noise) can lead to a better accuracy [8, 9], it would be impractical to force all users of a DNN inferencing platform to train their models with quantization from scratch. Instead, pre-trained floating point weights are directly quantized into MSFP, either offline or online within the hardware accelerator. Activation tensors are converted to MSFP in-situ on the hardware. At ultra-narrow bit-width, we found that a few steps (as low as 1 epoch) of model fine-tuning can help improve the accuracy of the quantized model. To fine-tune the network, we use the conventional training procedure based on stochastic gradient descent. During the forward pass, weights and activations are quantized to MSFP and the loss is calculated with the introduced quantization errors. A straight-through estimator in the back-propagation was to used for the gradients of quantization operators. The learning rate should be small during this fine-tuning phase, typically equal to the final learning rate used for original Float32 training.

MSFP was deployed on project Brainwave [10, 11], a datacenter-scale DNN inferencing service using networked FPGAs. Each Brainwave FPGA is a self-contained neural processing unit that has custom tensor-cores. Although the underlying configuration (i.e., bounding-box size and shape) can be re-configured based on the application requirements, we opt to use a fixed configuration in our experiments for simplicity (see Section 4). We emphasize that all data conversions and computation of shared exponent are handled in-situ on hardware without requiring users to do any pre-processing.

## 4 Experiments

To measure accuracy, the impact of MSFP on DNN inference was modeled in both Tensorflow and Pytorch using a custom library. This MSFP library was validated against our hardware for high fidelity emulation. For image classification, the majority of experiments focused on the ImageNet benchmark. The models used include ResNet-50 [12], ResNet-101, ResNet-152, Inception-v3 [13], Inception-v4 [14], MobileNet_V1_1.0_224 [15, 16], VGG16 [17], VGG19, and EfficientNet-EdgeTPU (-S, -M, and -L) [18, 19]. For transformer-based models, a pre-trained BERT-base [1] was used, and the accuracy of MSFP was evaluated on two downstream tasks: SQuAD (question answering) and MRPC (paraphrase detection). In addition, two proprietary RNN-based models named Production-DS and Production-DR were tested. Production-DS is a search relevance model that includes four GRUs with state size 500. Production-DR is a machine reading comprehension model that includes eight LSTMs with hidden dimension 100. Conversion to MSFP was applied to computation extensive layers as discussed in Section 2. Element-wise scalar operations such as activation functions were performed in Float16. A bounding box size of 16 was used unless otherwise specified. CNN models are typically used as backbone feature extractor in accelerated cloud-based applications. As such,

in those benchmarks, MSFP was applied to main backbone convolution layers only and the last fully-connected layer was kept in Float32. The last layer is usually run on commodity hardware and not the custom accelerator.

Table 2: Normalized accuracy of different benchmarks across a range of bit-widths. The values are normalized with respect to the Float32 model accuracy listed in column 3 in parenthesis. Configurations with the lowest bit-width that stays within 1% of Float32 accuracy are shown in bold.

| Category | Model | Float32 | MSFP16 | MSFP15 | MSFP14 | MSFP13 | MSFP12 |
|---|---|---|---|---|---|---|---|
| CNNs | Resnet-50 | 1.000 (75.26) | 1.000 | 0.999 | 0.994 | **0.989** | 0.967 |
| | Resnet-101 | 1.000 (76.21) | 1.000 | 1.000 | 0.998 | **0.991** | 0.964 |
| | Resnet-152 | 1.000 (76.58) | 1.000 | 1.001 | 0.997 | **0.991** | 0.968 |
| | Inception-v3 | 1.000 (77.98) | 1.000 | 1.005 | 1.001 | **0.990** | 0.943 |
| | Inception-v4 | 1.000 (80.18) | 1.000 | 1.001 | 1.000 | **0.993** | 0.963 |
| | MobileNet-V1 | 1.000 (70.90) | 0.998 | 0.997 | **0.990** | 0.965 | 0.863 |
| | VGG16 | 1.000 (70.93) | 1.000 | 1.004 | 1.005 | 1.003 | **1.002** |
| | VGG19 | 1.000 (71.02) | 1.000 | 1.002 | 1.001 | 1.002 | **1.000** |
| | EfficientNet-S | 1.000 (77.61) | 1.000 | 0.998 | **0.992** | 0.979 | 0.949 |
| | EfficientNet-M | 1.000 (78.98) | 1.000 | 0.998 | **0.993** | 0.980 | 0.950 |
| | EfficientNet-L | 1.000 (80.47) | 1.000 | 0.999 | **0.993** | 0.974 | 0.945 |
| RNNs | Production-DR | 1.000 (76.10) | 1.000 | 1.008 | 1.003 | 1.009 | **1.000** |
| | Production-DS | 1.000 (73.10) | 1.000 | 1.012 | 1.005 | 1.022 | **0.992** |
| Transformers | BERT-MRPC | 1.000 (88.39) | 1.000 | 1.005 | 1.002 | 1.008 | **1.018** |
| | BERT-SQuAD1.1 | 1.000 (88.45) | 1.000 | 0.998 | 0.998 | 0.997 | **0.990** |
| | BERT-SQuADv2 | 1.000 (77.23) | 1.000 | 0.999 | 0.999 | **0.993** | 0.989 |
| Circuitry | Memory density | 1.0× | 3.8× | 4.3× | 4.9× | 5.8× | 7.1× |
| | Arithmetic density | 1.0× | 8.8× | 10.8× | 13.9× | 18.3× | 31.9× |

Table 2 shows the normalized accuracy for a variety of different models. The accuracy values are normalized with respect to the Float32 counterpart. For image classification models, the metric is top-1 accuracy. Production-DS is evaluated using an area-under-curve (AUC) metric, BERT-MRPC is a classification task based on classic accuracy metric, and Production-DR and BERT-SQuAD use F1 score [20]. MSFP16 enables instant quantization while preserving the Float32 accuracy across various benchmarks without any fine-tuning or ad-hoc optimizations such as clipping or calibration. Model fine-tuning further enables pushing down the required bit-width for different benchmarks. Note that the quantization fine-tuning step is fairly low overhead (1-10 epochs) without any hyper-parameter tuning. We used the same learning rate as the final stage of original Float32 training for fine-tuning.

As shown in Table 2, CNN-based models typically require higher bit-widths to stay within 1% of the floating point result compared to RNNs and transformers. Note that even for CNN-based models one can drop the bit-width for weights to MSFP12 and still maintain high accuracy with MSFP as long as the activations are computed with MSFP13 or higher. Table 3 shows the results for using different bit-widths for weights and activations for ResNet-50 benchmark. We use a bounding-box size of 128 in this experiment which yields even better arithmetic density compared to default bounding-box size of 16. In general, we've observed that using more bits for activations produces higher accuracy than using more bits for weights. This paper focuses on uniform quantization. Mixed precision inference [21] with MSFP is an interesting extension for future work.

Table 3: Normalized accuracy of MSFP ResNet-50 for different weights and activation bit-widths. We use a bounding-box size of 128 in this experiment.

| Weight format | Activation format | | | |
|---|---|---|---|---|
| | MSFP15 | MSFP14 | MSFP13 | MSFP12 |
| **MSFP15** | 0.999 | 0.994 | 0.985 | 0.959 |
| **MSFP14** | 0.996 | 0.994 | 0.985 | 0.959 |
| **MSFP13** | 0.994 | 0.991 | 0.983 | 0.956 |
| **MSFP12** | 0.988 | 0.983 | 0.972 | 0.948 |

Figure 6 compares the accuracy versus relative multiplier density for different numerical formats. As shown, MSFP has superior performance while delivering high accuracy across various benchmarks.

We would like to emphasize that no special optimization (such as manual clipping, extensive hyper-parameter tuning, or quantization-aware training from scratch) is applied to boost accuracy. The same recipe is applied to all computer vision and natural language processing benchmarks.

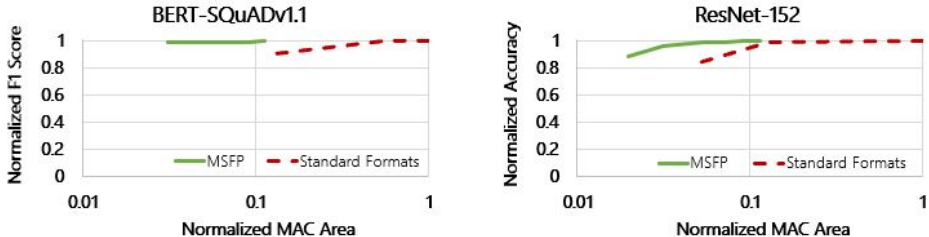

Figure 6: MSFP Pareto frontier (accuracy versus hardware overhead) compared to standard datatypes such as Float16, Int8, and Int4.

## 5 Related work

DNN inference with low bit-width arithmetic involves mapping a continuous set of values onto a discrete lattice (a.k.a., quantization). Over the past few years, a large body of work has looked into quantized DNN inference with fixed-point format [22, 23, 24, 25, 26, 7, 27, 28]. Outliers and irregular distributions, however, create a challenge for fixed-point quantization [29, 27, 30]. On the one hand, a uniform fixed-point quantization scheme that allocates lattice points for outliers will have fewer points available for dense portions of the distribution yielding large errors even though it incurs a low circuit footprint on custom silicon. On the other hand, focusing the quantization lattice on the dense portions of the distribution via a code-book [31] (a.k.a., non-uniform quantization) or posit [32, 33] datatype can potentially reduce quantization error but require radical changes to the arithmetic circuitry that are unlikely to be adopted for industry-scale deployment.

Finding the right balance between accuracy and hardware complexity is an active area of research. Despite several promising results on a few benchmarks [7, 30], the use of fixed-point arithmetic is considered a high-friction choice for large scale production, as it usually requires significant developer investment, hyper-parameter tuning, and/or re-calibration to adapt the proposed technique to a new domain. NVIDIA recently introduced a new software tool (called TensorRT) to adaptively transform an input Float32 model into int8 [34]. TensorRT requires model re-calibration and manual adjustments such as clipping to regain accuracy across various benchmarks. Quantizing below 8-bits with integer format often results in significant accuracy drop without significant model re-training.

MSFP addresses the aforementioned challenges by dividing the values in a tensor into fine-grained regions (bounding boxes) to limit the effect of outliers and irregularly distributed values. By independently quantizing subsets of a tensor, the probability that any given bounding-box suffers from outlier effects decreases. Compared to uniform and non-uniform quantization, MSFP is a hybrid quantization approach. Within each bounding region, a locally scaled uniform quantization is deployed to capture the distribution of each subset. The scaling factors used for different bounding regions, however, are independent of one another; resembling the structure of a non-uniform quantization. This hybrid approach lets us use a low overhead uniform lattice for each localized bounding-box with a low hardware cost while preserving a high level of accuracy by better taking care of local irregularities.

The idea of using shared exponents to enable computing with a precision approaching floating point while using a fixed-point processor [35, 36, 37] is not new in computer architecture or statistics. The pertinent parameter state space for DNN workloads, however, is relatively large and needs its own study. In this work, we carefully characterized the state space for different operations involved in DNN workloads. Our study led to identifying a set of parameterization that works robustly across various types of neural networks. We further provide an accompanying extensible ISA to process DNN graphs and map pertinent operations to custom accelerators without requiring users to do any kind of pre-processing. Our hardware/algorithm co-designed approach lets us significantly drive down the cost of DNN inference. Through extensive evaluations, we demonstrate the potential of

MSFP in the narrow bit-width regime. MSFP pushes the limits of narrow precision inferencing and enables operating at a new accuracy-cost Pareto frontier.

Our use of a shared exponent in MSFP shares similarity to FlexPoint [7]. FlexPoint, however, uses coarse-grained scaling factors (at the granularity of an entire layer) based on tensor-wide statistics which can lead to a significant accuracy drop (in the presence of outliers) or a high-friction pass to re-calibrate for new benchmarks [38]. In this paper, we show that using fine-grained scaling is the key to enable high performance narrow-precision quantization. In addition, unlike FlexPoint, all conversions to MSFP (including the selection of shared exponent) are automatically handled on the fly in-situ on our custom hardware without requiring pre-processing of activations/weights.

We emphasize that the primary objective of designing MSFP is accommodating a higher arithmetic density on a single node and not necessarily memory compression. While compression techniques such as pruning [31, 39, 40, 41], knowledge distillation [42, 43], weight sharing [44], or hashing [45, 46] result in promising memory compression, they may not necessarily yield a better arithmetic density and thus higher throughput and lower inference cost. In addition, this line of works requires changes to the topology of the pertinent model and usually incurs a low development velocity and a high re-training cost to obtain the original accuracy.

## 6    Conclusion

We presented Microsoft floating point as a robust datatype for low-cost DNN inference. Variants of MSFP together enable operating at a new accuracy-cost Pareto frontier compared to a collection of current standards. MSFP-based inference eliminates the need for extensive model re-calibration or ad-hoc optimization to preserve target accuracy. MSFP has the capability to effectively encode a wide range of tensors across different application domains including vision and language models. MSFP is deployed at a datacenter-scale and has been used to successfully ship over a dozen models.

## 7    Broader impact

This paper opens a new axis for the growing research in quantized DNN inference. It challenges the current practice in the field regarding the choice of numerical format and sheds light on the importance of a holistic co-design of hardware architecture and algorithms. This paper further highlights the importance of generalization in designing next generation standard quantization techniques to minimize the non-recurring engineering cost and ensure ease-of-use across various classes of models.

Large-scale cost-efficient inference over DNNs enables an ever-increasing number of AI applications in consumer and enterprise products. MSFP enables inferencing on larger and more powerful DNN models in scenarios that require very high rates of inference such as web search, enterprise search, and email search. Other scenarios such as real-time recommendations, AI-powered text auto-completion (e.g. auto-suggestion, smart compose), and conversational interfaces also require high inference rates and benefit from inferencing with MSFP format.

### Acknowledgments and Disclosure of Funding

We thank the anonymous reviewers and program committee for their feedback.

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
