[Reviews · NeurIPS 2020]

Review 1

Summary and Contributions: This paper shows a grouped numerical format that represents contiguous values in tensors along some dimension with a shared exponent, similar to traditional block floating point / Flexpoint (as the authors mention), except with controllable granularity and direction of the sharing. Since a group of floating point values being operated upon share the same exponent, this avoids issues with significand alignment for summation (for inner product) and allows for a tradeoff between encoding the dynamic range and the significand precision. Given relatively high dynamic range (8 bits of base-2 exponent here), the authors find that in typical application the significand widths can be decreased significantly, which leads to quadratically smaller multiplication hardware and linearly smaller adders. The overall result is that a BBFP processing element/ALU can be much higher density / lower power (and probably lower latency) than the alternatives. While the choice of how the bounding box structure is applied to linear algebra computations is somewhat up in the air with this paper, the authors show that when applied to a wide range of NN inferencing tasks, performance is quite acceptable, and with minimal fine tuning or other adjustments to the network. Generall;y this work seems applicable to low-power or high-speed inference due to this increase in density, and seems superior to integer quantization methods in this regard.

Strengths: I believe this work provides a novel way to trade off dynamic range with significand precision that is part way between the Flexpoint approach and the prefix-free encoding approach within single scalar values such as the posit design. The Flexpoint approach similarly can be applied to reduce multiplier and adder sizes, but appears to suffer too many problems due to the granularity of the shared exponent. While the posit-style approach makes better use of the bits for matching the data distribution of encoded scalar values (a higher entropy encoding than typical fixed-width fields), but the circuits required are more complicated and only increase adder and multiplier sizes. While memory reduction was not the primary goal of the authors, this approach also has the side benefit of a reduction in needed bandwidth to the circuitry due to a 4-7x reduction in encodings. The greatest benefit is in the ALU sizes especially for inner product, and the advantage seems pretty clear to me. This approach is another technique that is applicable for developing either high throughput or lower power NN inferencing hardware.

Weaknesses: Some of these I'm listing here in weaknesses are more a lack of clarity in how this arithmetic works than things which may be weaknesses per se. Can zero be represented at all in this format? Typical floating point formats presume an implicit leading integer 1 for the significand (with the remainder of the bits being the fractional part of the significand, and the exception being IEEE 754 subnormal encoding which gives an implicit leading integer 0 for the significand). Presuming IEEE-style usage, zero can only be represented if all values in the bounding box are zero (the shared exponent is zero, and all significand bits are zero). Or is zero approximated by the smallest representable value within the shared exponent. How would, say, ReLU be implemented then? It is unclear how the bounding box shapes or how they are aligned/tiled is chosen. Is it ever the case that you have operations between mismatched bounding box sizes due to different tiling patterns, or must the direction of inner product always be tiled in the same way. How are odd sizes handled (a trailing bounding box with ignored remaining values), which may be produced from certain convolution kernel sizes or pooling sizes. This is not really described at all save for a cursory comment in Figure 3. How would, say, strided or à trous convolution be implemented? Is this done by a separate scatter/gather engine on top of this to restructure data, or are these operations possible at all? I understand how inner product would work, but how does pointwise addition/subtraction work (e.g., as seen in a residual network)? Is significand alignment performed for all values based on exponent difference? What happens in the case of carries (some scalar values want to increment the exponent, while others do not, or a subtraction wants to decrement the exponent)? It seems that when forming a bounding box even for the results of many inner products (e.g., C = AB, the scalar entries of C are the results of multiple BBFP inner products), what happens when you attempt to repackage C into new bounding boxes with a shared exponent. How would, e.g., transcendental functions like sigmoid, log or exp be implemented using this format and this hardware design? If these need to be farmed out to separate hardware that did not maintain the bounding box approach, then this would penalize network designs that frequently had such things. What hardware overhead is needed for maintaining the extent of bounding boxes, or the need to repackage single scalar results (e.g., the scalar entries of C as results of multiple BBFP inner products in C = AB above)?

Correctness: While it is unclear to me how certain aspects of the arithmetic and the tiling work in practice, it appears that the authors have tested a fair number of architectures and thus likely deal with some of the things that I list as weaknesses in my mind, and the technique seems generally applicable. I do feel like I have to take it on trust that the method does work though based on the results. The circuit design efficiencies however are generally obvious and quite large in my mind, though.

Clarity: Some of this I covered in the weaknesses section, but I think it is unclear how some aspects of the arithmetic work (zero representation, pointwise addition/subtraction). The description of BBFP on page 3 itself was a little confusing to me when I first encountered it. Following: "the number listed indicates the sum of the bit-width assigned to sign bit, mantissa bits, and shared exponent" followed by "we fixed the number of exponent bits to 8". Putting together those two statements, my initial interpretation was that a single BBFP12 word is 12 bits, which consists of an 8 bit shared exponent, leaving 4 for remaining sign bits and mantissas. This is divisible by 2, so BBFP12 contains likely two values, each with a single sign bit and single mantissa bit? But then this made no sense, as then what does BBFP15 or BBFP13 or BBFP11 represent, since those minus 8 (shared exponent width) are prime numbers, so cannot contain more than a single value? Only later on did I seem to get that you mean that each individual member of a bounding box for BBFP12 is (12 - 8) = 4 bits, thus a single sign bit, 3 mantissa bits, so a bounding box containing 4 elements would thus be 8 + 4 * 4 = 24 bits in size. This could probably be made clearer in the initial description. Otherwise, I felt like the paper was fairly well written, if vague in many areas.

Relation to Prior Work: I think section 5 covers this fairly well. With regards to inference, there are other techniques that might be worth mentioning. The product quantization-like technique in And The Bit Goes Down: Revisiting the Quantization of Neural Networks (Stock et al., ICLR 2020) might be worth discussing, as it is to my mind the state of the art for non-sparsification or pruning based quantization. I understand though that the target of that work is not the same as the target of this work (which is to increase arithmetic efficiency). For memory compression, though I think Stock et al. will not map to efficient circuits due to excessive LUT usage and the fact that it doesn't quite address arithmetic demands.

Reproducibility: No

Additional Feedback:


Review 2

Summary and Contributions: The paper proposes a variant of blocking floating point numerical format (rebranded as Bounding-Box Floating-Point (BBFP)). In contrast with previous works, the window's size (bounding-box size) of the shared exponent is parameterized. The inference performance of deep neural networks, recurrent neural networks, and transfer models using this numerical format are evaluated and compared with a 32-bit floating-point format. Moreover, the arithmetic density of this numerical format and other numerical formats such as Bfloat16 and INT8 and INT4 are discussed.

Strengths: 1- The proposed numerical format is evaluated in large and variant of benchmarks which mean the proposed numerical format can be deployed for different models 2- Presenting the relationship between the gaussian quantization noise and the Bounding-Box Floating-Point quantization noise in Table 2 is interesting.

Weaknesses: 1- Is the QNSR defined by the power of a signal (x^2) and noise (N^2) rather than signal and noise itself? I also suggest using QSNR to ignore the negative value, which is more readable. 2- This new numerical format is compared with FP32 across various benchmarks. Compared with different numerical formats such as adaptive float [1], posit, and uniform and block floating points in the same number of bits are missed. [1] Tambe, Thierry, et al. "AdaptivFloat: A Floating-Point Based Data Type for Resilient Deep Learning Inference." arXiv preprint arXiv:1909.13271 (2019).

Correctness: The author claims that the other numerical formats like INT8 require the re-calibration to show promise results, and BBFP is not needed. This claim is not correct. The BBFP also requires an algorithm to find the window size based on application automatically. The KL algorithm used to find the windows' size is similar to the calibration algorithm in TensorRT. The overhead of using BBFP compared to other numerical formats such as INT8, INT4, Bfloat16, Float8 [1], and Float4 [1] should be discussed clearly in the paper.

Clarity: Figure 1 is unclear. How is the energy and cost of MAC with INT8 is more than BBFP12? How the MAC energy and cost is calculated in figure 1? How is accuracy calculated? Is INT8 combined with linear quantization?

Relation to Prior Work: Yes

Reproducibility: Yes

Additional Feedback: ==== Update after author feedback ==== The author addressed most of my review comments. The explanation of hardware design and hardware analysis needs to be improved. Moreover, the overhead of memory and the large size of the shared exponent are still issues, which are not addressed in the rebuttal.


Review 3

Summary and Contributions: This paper describes a technique for efficient quantization of pre-trained DNN weights and activations. Typical floating point representations comprise a sign bit, an exponent and a mantissa. The key idea of this paper is to share the exponent across a group of elements (a so-called bounding box). This scheme reduces both the number of bits required to represent the numbers in the box as well as the arithmetic required for dot product operations. If the boxes are designed so that elements in the same box have roughly the same scale, then accuracy of the resulting DNN classification will not be adversely affected. The paper describes the proposed representation (BBFP) in a lot of detail, including how they design the size and shape of the bounding box, as well as how they perform fine-tuning to improve the accuracy when the number of mantissa bits is made extremely small. Finally, the paper presents a lot of experimental results of different neural networks (CCNs, RNNs and Transformers) to quantify the loss in accuracy relative to the gains in terms of memory and arithmetic density.

Strengths: DNN models are becoming widely used in production systems and many different groups are developing custom hardware to make such systems faster and/or more energy efficient. The custom floating point format proposed in this paper can be used to reduce the area and energy footprint of DNNs and thus is of significant interest to the NeurIPS community. The BBFP scheme is clearly described mathematically and the experimental improvements over FP32 in terms of memory and arithmetic density at the same accuracy level are impressive and significant.

Weaknesses: Very little space is dedicated to describing the fine tuning part of the pipeline. In particular, it was not clear to me what using a "straight-through estimator for the BBFP quantization" during the backward pass means. It would also improve the paper to provide some quantitative comparison of this fine-tuning effort vs. whatever is needed to get fixed-point schemes (e.g. INT8) working well. My other main concern is novelty. The authors indicate that the idea of sharing the exponent across blocks of elements is not, by itself, novel. Previously work (Flexpoint) has studied the effect of sharing the exponent across entire tensors. The authors argue that this is too coarse an approach and that the accuracy suffers accordingly. In that sense, the main novelty of BBFP seems to be how the bounding boxes are defined which seems to be performed in a relatively heuristic manner (e.g. Figure 3). In Section 5 the authors state that another difference between their solution and Flexpoint is that "all conversions to BBFP are automatically handled on the fly in-situ on our custom hardware". It might add to the novelty if these on-the-fly conversions were described somewhere in the paper.

Correctness: I do not see anything that is obviously incorrect in the BBFP method or the experiments. However, I think more details regarding the fine tuning are needed (see above) and at least one more baseline (Flexpoint) is needed in the experimental results.

Clarity: The paper is written to a very high standard and reads well.

Relation to Prior Work: The work is placed clearly in the context of the existing work and I feel the discussion is adequate. However, the paper would be stronger if the authors also included the closest similar approach (Flexpoint or some INT8 scheme) in the experimental comparison (e.g. Table 3).

Reproducibility: Yes

Additional Feedback: ==== Update after author feedback ==== I thank the authors for clarification regarding the comparison with Flexpoint + INT8, as well as the fine tuning. I've updated my score accordingly.


Review 4

Summary and Contributions: This paper proposes a new floating-point format for neural network inference. The proposed format shares the exponent per each chunk of several numbers. Contributions include: 1. the proposed numerical format and some empirical study on the selection of hyperparameters; 2. a co-designed hardware (?) with less MAC area; 3. accuracy results on a wide variety of models.

Strengths: This paper addresses an important problem of accelerate the inference of neural networks. The idea of sharing exponent per chunk is theoretically sound. The experimental result is strong. It shows a significantly reduction of memory and arithmetic density, under minor accuracy loss, on a wide variety of tasks.

Weaknesses: I think the biggest problem of this paper is reproducibility. I think the most attractive point of this paper is the hardware part, which leads to a significant lower cost. However, except the results in Table 1&3, I didn't find any text describing the design of the hardware, how to implement matrix multiplication with the proposed format on the hardware, and why the proposed numerical format leads to so much cost reduction. Thus this paper looks incomplete from my perspective. Furthermore, the algorithm part also misses some details, for example, how is the KL divergence in Fig. 2 defined? What is the bounding box side along each spatial dimension (batch size, channel, height, width)?

Correctness: Yes

Clarity: It is mostly clear and easy to understand. However I feel the paper is incomplete, as mentioned before.

Relation to Prior Work: Yes.

Reproducibility: No

Additional Feedback: Post rebuttal ==== My concerns are addressed in the authors' response. Therefore I increased my score.

[Author Response · NeurIPS 2020]

We thank the reviewers for their appreciation of our work and helpful feedback. Here, we address open questions.

**[Reviewer 7] Reproducibility of hardware**: Following figure shows a systolic tensor core architecture containing
multiple BBFP dot product units. Each dot product unit has a significantly lower circuit footprint compared to
conventional float due to the shared exponent. The math per bounding-box is mostly performed in fixed-point format
and the cost of dynamic scaling is amortized over $n$ (block size). We'll amend the camera-ready with more details.

BBFP provides 3x and 4x higher MAC density compared to industry
standards Bfloat16 and INT8, respectively

**[Reviewer 7 and 1] Conversion to BBFP**: the axis along which exponent is shared is always the inner dimensions
of a matrix multiply: for $\mathbf{A} \times \mathbf{B}$ we have shared exponents along the rows of $\mathbf{A}$ and the columns of $\mathbf{B}$. If the inner
dimension is not dividable by the bounding box size, we pad the last bounding box with zero.

**[Reviewer 3 and 2] Comparison with FlexPoint, INT8, INT4, and AdaptivFloat**: As discussed in the paper,
leveraging shared exponent to lower hardware cost is not a new idea. However, there is a very large search space of
designs across bit-width, exponent selection policy, block granularity, bounding box policy—coupled with hardware
implementation—that makes BBFP non-obvious. In this paper, we showed how using a balanced fine-grained approach
can provide a robust recipe that works across various models. Due to the low user-friction of BBFP format and its
high performance, this datatype has been adopted by different teams and deployed in large-scale production. FlexPoint
coarse-grained approach, however, results in significant accuracy drops (in presence of outliers) and incur a high-friction
pass to recover accuracy. We evaluated the accuracy versus MAC density trade-off for different benchmarks in Figures 1
and 5. This comparison includes industry-standard datatypes (bfloat16, INT8, and INT4). Different datatypes' circuitry
is compared in Table 1. We'll append Table 1 to include float8, float4, and posits in the revision. Overall, float8
has $1.57\times$ and $3.46\times$ more area overhead compared to BBFP16 and BBFP12, respectively. As for comparison with
AdaptiveFloat, in ResNet50-ImageNet with an average of 4 bits (sign plus mantissa), BBFP preserves $96.7\%$ of the
baseline accuracy whereas AdaptivFloat preserves $95.2\%$. Also, BBFP uses a uniform bit-width for all layers whereas
AdaptivFloat adjusts the bit-width per layer, making it more complicated on hardware. Finally, AdaptivFloat reports up
to $1.14\times$ area improvement compared to INT8 whereas BBFP has a $4\times$ lower overhead.

**[Reviewer 3 and 2] Fine-tuning process and calibration**: Quantization involves discretization processes such as
rounding and truncation that result in null gradients. The idea of straight-through estimator is to replace the gradient of
those operators with identity matrix. We'll elaborate more in the revision. Integer-based inference requires calibration
due to their fixed dynamic range—tensors must be scaled to the proper range to be represented. BBFP does not require
this type of calibration as it already has a scaling exponent. As for bounding-box selection, all experiments in section 4
including Tables 3, 4, and Figure 5 have been performed with a fixed bounding box size of 16 (no calibration involved).

**[Reviewer 2] Elaboration on Figure 1**: BBFP12 has 4-bit sign-magnitude mantissas and an 8-bit shared exponent.
BBFP12 dot-product circuitry is comparable to INT4. Unlike BBFP, INT format follows a two's complement represen-
tation. Two's complement MAC costs more in area/energy compared to a sign-magnitude MAC of the same bit-width.
We'll make the notation more explicit in the revision. MAC energy/area cost are measured based on a topographical
synthesis of these MAC units on TSMC 16nm FF+. Power simulations are based on an input toggle rate of 50% with
50% static probability @1GHz. Figure 1 accuracy numbers are for Resnet50-ImageNet.

**[Reviewer 1] Non-matmul operations and support for strided and atrous convolution**: BBFP is designed to
improve dot product performance. All vector operations such as sigmoid, activation functions, point-wise addition are
performed in-situ on HW in float16 (with conversions to BBFP supported automatically on HW). Any operation that
can be broken down into dot products (including strided/dilated convs) is supported by BBFP. Scatter/gather is one way
of reshaping the input data for strided/dilated convs (SRAM address striding and crossbar are other alternatives).

**[Reviewer 1] Zero representation**: Zero is represented by having all mantissa bits being 0 for a given value (shared
exponent can be any value). BBFP mantissas do not have an implicit leading bit and all mantissa bits are explicitly
represented. BBFP does not have a representation for NaN/Inf, but this does not impact DNN inference accuracy.

**[Reviewer 7 and 2] KL Divergence and QNSR**: The KL divergence from BBFP to float32 is computed after computing
the pertinent normalized histogram of values in each encoding format. KL divergence is defined as $KL(P \parallel Q) =$
$\sum_{x \in \mathcal{X}} P(x) \log\left(\frac{P(x)}{Q(x)}\right)$. We will replace QNSR with QSNR in the revised paper to avoid negative values.

[Meta-Review · NeurIPS 2020]

Four knowledgeable referees support acceptance for the contribution mainly due to its novelty and practical impact, and I also recommend acceptance. The only concern is about clarity and reproducibility; Most of the uncertain parts raised by reviewers have been resolved through rebuttal, but I ask authors to thoroughly go through the paper so that there is no lack of explanation in all areas of the final version.